# Mitochondria Turnover and Lysosomal Function in Hematopoietic Stem Cell Metabolism

**DOI:** 10.3390/ijms22094627

**Published:** 2021-04-28

**Authors:** Makiko Mochizuki-Kashio, Hiroko Shiozaki, Toshio Suda, Ayako Nakamura-Ishizu

**Affiliations:** 1Microanatomy and Developmental Biology, Tokyo Women’s Medical University, 8-1 Kawadacho, Shinjuku-ku, Tokyo 162-8666, Japan; mochizuki.makiko@twmu.ac.jp; 2Department of Hematology, Tokyo Women’s Medical University, 8-1 Kawadacho, Shinjuku-ku, Tokyo 162-8666, Japan; hshiozaki@cpost.plala.or.jp; 3Cancer Science Institute, National University of Singapore, 14 Medical Drive, MD6, Singapore 117599, Singapore; sudato@keio.jp; 4International Research Center for Medical Sciences, Kumamoto University, 2-2-1 Honjo, Chuo-ku, Kumamoto City 860-0811, Japan

**Keywords:** hematopoietic stem cells, ROS, mitochondria, lysosome, autophagy, folliculin

## Abstract

Hematopoietic stem cells (HSCs) reside in a hypoxic microenvironment that enables glycolysis-fueled metabolism and reduces oxidative stress. Nonetheless, metabolic regulation in organelles such as the mitochondria and lysosomes as well as autophagic processes have been implicated as essential for the determination of HSC cell fate. This review encompasses the current understanding of anaerobic metabolism in HSCs as well as the emerging roles of mitochondrial metabolism and lysosomal regulation for hematopoietic homeostasis.

## 1. Introduction

Hematopoietic cells consist of a heterogenous group of cells originating from hematopoietic stem cells (HSCs) [1]. HSCs differentiate into multi-potent progenitor cells (MPPs) which further produce circulating and tissue-residing blood cells of specific lineage [2]. The bone marrow (BM) is the main hematopoietic organ in an adult and houses millions of immature and mature hematopoietic cells. HSCs reside as a rare cell population in the BM where they are maintained in quiescence as a reserve pool for hematopoiesis [3]. When hematopoiesis is compromised, HSCs self-renew, proliferate and differentiate to replenish hematopoietic cells.

As with all cells in an organism, hematopoietic cells utilize adenosine triphosphate (ATP) as a common energy currency. While it is fundamentally known that ATP is produced anaerobically through glycolysis and aerobically through mitochondrial oxidative phosphorylation (OXPHOS), the contribution of these processes for energy production varies between hematopoietic cell type. While the BM microenvironment is predominantly hypoxic [4], different hematopoietic cells utilize both glycolysis and OXPHOS at varying degrees for survival. Here we review the metabolic state and regulation of murine HSCs, particularly emphasizing non-oxygen- and oxygen-associated processes in HSC regulation. We will also highlight folliculin (FLCN)-Transcription Factor Binding To IGHM Enhancer 3 (TFE3)/Transcription Factor EB (TFEB) pathway as a critical metabolic regulator of mitochondria and lysosomes in HSCs.

## 2. Glycolysis and Oxidative Phosphorylation (OXPHOS) during Hematopoietic Differentiation

Stem cells rely on glycolysis for energy production in order to reduce oxidative stress, yet upon differentiation, require a robust upregulation of mitochondrial OXPHOS [5]. Likewise, HSCs require a shift in anaerobic to aerobic metabolism to efficiently proliferate and produce mature hematopoietic cells (hypoxic regulation of HSCs has been reviewed extensively in [6]). HSCs reside in the BM niche that is hypoxic compared to other tissues [4,7]. The BM hypoxic niche influences HSCs to produce low amounts of energy via anaerobic glycolysis. HSCs express high levels of hypoxia inducible factor-1 alpha (HIF1α) and Hif1α deletion results in loss of HSC quiescence and stem cell potential [6,8]. The deletion of Meis Homeobox 1 (Meis1), which activates transcription of Hif1α, also results in a similar loss of function in HSCs [9,10]. Remaining in a state of metabolic quiescence, HSCs withstand premature exhaustion from cell proliferation. In association, hypoxic culture conditions maintain HSCs in quiescence ex vivo [11]. Specifically, hypoxic culture conditions (in 1% O_2_ conditions), low concentrations of stem cell factor (SCF) and thrombopoietin (Thpo) with the addition of 4% bovine serum albumin (BSA) as a source of fatty acids maintained quiescent murine HSCs for 1 month in culture [11]. While another recent protocol succeeded in optimizing ex vivo expansion of HSCs under normal oxygen pressure [12], hypoxic culture characteristically maintained HSCs in quiescence.

Adult HSCs depend upon discrete genetic and proteomic control in order to utilize anaerobic glycolysis as their main energy source [8,13]. HSCs exhibit high expression of glycolytic enzyme, pyruvate dehydrogenase kinase (Pdk), which downregulates aerobic metabolism through inhibition of pyruvate dehydrogenase (PDH)-mediated conversion of pyruvate to acetyl-CoA [14]. Loss of Pdk2 and Pdk4 in HSCs results in loss of stem cell potential. Furthermore, BM cells from mice deficient of glycolytic enzyme, lactate dehydrogenase A (Ldha) or M2 pyruvate kinase isoform (Pkm2) exhibit significantly low repopulation potential upon BM transplantation [15]. Interestingly, mitochondria activity and reactive oxygen species (ROS) production increased in HSCs only with Ldha and not with Pkm2 deficiency suggesting a complex regulatory network between anaerobic and aerobic metabolism in HSCs.

Mitochondria contain the major enzymes that oxidize carbohydrates, proteins and lipids to produce ATP [16]. In this process, appropriate substrates are catabolized to acetyl-CoA which enters the TCA cycle (tricarboxylic acid cycle, or Krebs cycle) in the mitochondrial matrix to form reduced cofactors, NADH (nicotinamide adenine dinucleotide) and FADH2 (flavin adenine dinucleotide). NADH and FADH2 are fueled tothe electron transport complex (ETC) on the inner mitochondria membrane (IMM) for aerobic ATP production [17]. Although HSCs predominantly rely on glycolysis, mitochondrial metabolism is not completely dispensable for their survival. Molecular components of mitochondria OXPHOS have been implicated for the maintenance of HSCs. Deletion of an IMM-specific phosphatase (PTEN-like mitochondrial phosphatase, PTPMT1) resulted in defective aerobic mitochondria metabolism and BM failure [18]. Interestingly, deletion of *Ptpmt1* in mature hematopoietic lineage cells (*Ptpmt1fl/fl/LysM-Cre+* (deletion in granulocyte-macrophage lineage), *Ptpmt1fl/fl*/*Lck-Cre*+ (deletion in T cell lineage) and *Ptpmt1fl/fl/CD19-Cre+* (deletion in B cell lineage)) did not influence the number and function of these mature cells. Deletion of Rieske iron-sulfur protein (RISP) located in mitochondrial complex III of the ETC also impaired both fetal and adult hematopoiesis [19]. RISP-deficient fetal HSCs exhibit defective differentiation resulting in severe anemia. Induction of deletion in mice also results in a loss of quiescence and survival potential in adult HSCs. Importantly, deletion of RISP resulted in decreased NAD+/NADH ratio, which caused accumulation of 2-hydroxyglutrate (2-HG), an epigenetic moderator. Also, a small portion of DNA is located in the mitochondria (mtDNA) and its integrity influences HSC aging. HSCs with defective a proof-reading-deficient version of mitochondrial DNA polymerase gamma (Polg) exhibit premature aging phenotype [20]. Further investigation is necessary to characterize HSC mitochondria functions under hypoxic conditions and to identify specific metabolites of mitochondria which potentially serve to maintain HSCs.

Glycolysis and mitochondrial OXPHOS in HSCs are interlinked with other metabolic pathways such as glutamine metabolism. Glutamine metabolism, or glutaminolysis, provides energy for proliferative cells and involves the deamination of glutamine by glutaminase (Gls1) in the mitochondria. HSCs exhibit low rates in both glucose and glutamine consumption [21]. It was shown that alternative polyadenylation (APA) regulated conversion of Glsisoforms which resulted in upregulation of glutamine metabolism to fuel energy for HSC self-renewal [22]. Glucose and glutamine metabolism have also been implicated in HSC differentiation to erythroid lineage [23]. HSCs commit to erythroid lineage differentiation through high expression of the glutamine transporter, ASCT2 (SLC1A5), which stimulated glutamine-driven de novo nucleotide biosynthesis.

## 3. Reactive Oxygen Species (ROS) Production in Hematopoietic Stem Cells (HSCs)

ROS are a group of oxygen-containing molecules that easily interact with and cause oxidative damage to lipids, proteins, and nucleic acids [24]. The electron reduction of oxygen molecules forms superoxide anions (O_2_^−^), hydrogen peroxide (H_2_O_2_) and hydroxyl radicals (OH^−^), the major forms of ROS. Mitochondrial OXPHOS is the major source of cellular ROS production. Approximately ∼0.1–0.2% of oxygen consumed by mitochondria converts to ROS through the electron flow of oxygen in the ETC [25]. ROS is also produced through NADPH oxidase (NOX) mediated conversion of NADPH to NADP+. In order to reduce ROS-associated cellular damage, ROS production is minimized through the suppression of OXPHOS in HSCs [6,26]. Reflecting their low OXPHOS activation, HSCs exhibit low ROS levels compared to differentiated hematopoietic cells and HSCs with lower levels of ROS retain higher stem cell potential [27,28]. Other than minimizing OXPHOS activity, HSCs exhibit multiple mechanisms which reduce ROS [29,30,31,32,33]. Intracellular ROS accumulates when HSCs exit from quiescence and proliferate upon various stress [34]. In relation, HSCs exposed to brief periods of ambient oxygen showed a rise in OXPHOS activity and ROS levels, which lowered stem cell potential [35]. This phenomenon, named “extra physiologic oxygen shock/stress” (EPHOSS), thus emphasized that hypoxia and reduction of oxidative stress was crucial for ex vivo manipulation of HSCs.

Importantly, ROS levels can be reduced by fueling glucose metabolites away from the mitochondria into the pentose phosphate pathway (PPP). Upon entering the cell, glucose is converted to glucose-6-phosophate (G6P) which may be fueled to PPP which yields NADPH with reduced functions [36]. The activity of the key enzyme for PPP, glucose-6-phosphate dehydrogenase (G6PD) is regulated by a cytoplasmic NAD+-dependent deacetylase, sirtuin 2 (SIRT2), and is essential for the growth and survival of leukemic cells [37,38]. While the suppression of SIRT2 and G6PD using small molecule inhibitors did not inhibit growth of cultured hematopoietic stem and progenitor cells (HSPCs), an in vivo study demonstrated that Sirt2-deficient aged mice exhibit loss of HSC stem cell function [39]. Sirt2-deficiency was associated with nucleotide-binding oligomerization domain-like receptor family, pyrin domain-containing 3 (NLRP3) inflammasome activation and mitochondria stress in aged HSCs. While G6PD is essential for erythropoiesis [40], its activity and the role of PPP for energy and redox reaction in primary HSCs are yet to be analyzed.

## 4. Reduction of ROS through Redox Regulation in HSCs

HSCs have strategies to minimize the production of ROS to maintain the stemness. HSCs reserve multiple anti-oxidant processes which reduce ROS levels. One of them is by reducing ROS through the activity of reductive peptides, peroxiredoxins and antioxidant enzymes. Initially, the function of the major antioxidant enzyme superoxide dismutase (SOD) deficiency was directly implicated in erythropoiesis [41]. In the case of HSCs, deletion of forkhead box O transcription factor, Foxo3, resulted in an accumulation of ROS in HSCs due to reduced production of antioxidant enzymes, including catalase and superoxide dismutase 2 (SOD2) [42,43,44]. Deletion of Foxo3 thus causes loss of HSC quiescence and exhaustion. Glutathione-dependent enzymes catalyze biological redox reactions. Deficiency of microsomal glutathione transferase 1 (Mgst1) results in defective HSC differentiation [45]. The transcription factor, nuclear factor erythroid 2-related factor 2 (Nrf2), regulates multiple anti-oxidant processes and its role in hematopoiesis has been studied extensively [46]. *Nrf2^−/−^* HSCs lose quiescence and stem cell potential [47]. However, since stem cell phenotypes of *Nrf2^−/−^* HSCs cannot be restored by the anti-oxidant treatment with N-acetyl cysteine (NAC), non-redox roles of Nrf2 should be considered in HSC regulation [48].

## 5. The Effect of ROS on HSC Fate

ROS production seems to be a double-edged sword in HSC regulation; ROS acts as vital cellular signaling molecules yet also is detrimental to cells (Figure 1) [21]. Adequate levels of ROS promote HSC differentiation and ROS levels differ in hematopoietic cells of different lineage [49]. ROS functions as a cell signal molecule which provokes and activates downstream signaling pathways of various hematopoietic growth factors via tyrosine phosphorylation [50]. While considerable ROS levels stimulate vital processes, excessive accumulation is detrimental in a cell [34]. ROS is the main driver of DNA double strand breaks through the production of 8-oxo-2,-deoxyguanosine (8-oxodG) lesion in proliferating HSCs during inflammatory stress. ROS-induced DNA damage is also noted in cultured human umbilical cord stem and progenitor cells [51]. Moreover, ROS levels can influence DNA damage response directly or indirectly. Oxidative stress may induce nuclear co-localization of Foxo3a and DNA damage repair protein, Fanconi anemia group d2 (Fancd2) [52]. Double knock out of *Foxo3a* and *Fancd2* accelerated HSC exhaustion in mice [53]. These data indicate that ROS regulation and DNA damage repair mechanisms intertwine in a complex way and further investigations would be needed to elucidate the crosstalk.

ROS level also potentially affects HSC fate through modulation of the epigenetic landscape. Chromatin remodeling and dynamics heavily impact HSC stem cell fate as well as aging [54]. Hydroxyl radicals stimulate the conversion of 5-methylcytosine (5mC) to 5-hydroxymethylcytosine (5hmC), through aberrant transcriptional activation, and subsequently promote DNA demethylation [55,56]. Along these lines, epigenetic modifications depend upon products from mitochondrial oxidative metabolism. For example, the epigenetic regulator, ten-eleven translocation (TET) enzyme activity depends on 2-hydroxyglutarate (2-HG), a product of α-ketoglutarate (αKG) originating from the TCA cycle [54]. Glutaminolysis and catabolism of branched-chain amino acid (BCAA) also result in αKG production and these metabolic processes influence HSC and leukemic stem cell (LSC) regulation [57,58]. While the production of ROS and mitochondrial dysfunction clearly affects epigenetic modifications, further analysis, especially on the context of interrogating these processes during HSC aging, is needed.

## 6. Mitochondria Volume and Turnover in HSCs

Mitochondria are main organelles which produce ROS during the generation of ATP. Surprisingly, despite low OXPHOS activity and ROS production in HSCs, cellular mitochondrial volume is discrepantly high in HSCs [59,60]. Both detection of OXPHOS activity and mitochondrial volume have relied on dye-based methods such as TMRE (tetramethylrhodamine, ethyl ester) or MitoTracker staining [60,61]. Based on these methods, OXPHOS activation, quantified as mitochondrial membrane potential (ΔΨmt), is low in highly potent HSCs [19,62,63]. However, due to the presence of efflux pumps in HSCs, dye-based staining methods may underestimate the true values of mitochondria properties [64]. Accordingly, dye-based staining methods used in conjunction with various efflux pump inhibitors have yielded controversial results [60,61,65,66]. Interestingly, when mitochondrial volume was quantified through non-dye based methods such as mitochondrial DNA levels, the assessment of mito-dendra2 fluorescence or immunohistochemical staining of TOM20 (Translocase of outer membrane 20), HSCs exhibited higher mitochondrial volume compared to progenitor cells [60]. Furthermore, we reported that HSCs with higher mito-dendra2 fluorescence were quiescent and repopulated higher than HSCs with low mito-dendra2 fluorescence, which indicated that high mitochondrial volume parallels stem cell potential [59]. These results indicate that adult quiescent HSCs retain substantial numbers of mitochondria.

The fact that mitochondrial volume varies and associates with stem cell potential in HSCs necessitates further understanding of mitochondrial dynamics in HSCs (reviewed also in [67]). Mitochondria fission and fusion also influence the volume of mitochondria in a cell. Mitochondria fission regulator protein, Dynamin-related protein 1 (Drp1), was highlighted as a crucial factor which affects HSCs during stress induced replication [68]. Drp1-deficient HSCs exhibited asymmetric segregation of damaged mitochondria resulting in functional decline of HSCs upon BM transplantation. Another fission regulator protein, mitochondrial fission protein 1 (Fis1), regulates mitochondria activity and preserves self-renewal of LSCs proliferation but affects normal human HSPCs to a lesser extent [29].

Mitochondria turnover is regulated through mitochondrial biogenesis and clearance through autophagy. Autophagy activation is vital for the regulation of healthy HSCs. Autophagy-related genes, *Atg7*- and *Atg12*-deficient mice exhibit reduced numbers of HSC and progenitor cells [69,70]. Defective autophagy stimulated the accumulation of damaged mitochondria with a high ΔΨmt which subsequently increased mitochondrial oxidative stress and DNA damage. Similarly, the deletion of *Atg5* in hematopoietic cells causes defective clearance of ΔΨmt high mitochondria and results in low HSC reconstitution potential [71].

Mitochondrial clearance through autophagy, termed as mitophagy, eliminates dysfunctional mitochondria as a whole organelle and controls mitochondrial volume. Mitophagy in mammals is initiated through the recognition of mitophagy receptor proteins, such as BNIP3L/NIX, mitochondria-associated BH3-only protein (BNIP3), and FUN14 domain containing 1 (FUNDC1) on the mitochondria outer membrane by proteins with LC3-interacting region (LIR) on the autophagosomes [72]. These interactions allow autophagosomes to engulf mitochondria and subsequent fusion to lysosomes for degradation. The role of mitophagy in HSC regulation was highlighted through investigating Pten-induced putative kinase 1 (Pink1) interactions with the E3 ubiquitin ligase Parkin (or Park2) [73] (also reviewed in [74]). Pink1, a mitochondrial serine threonine kinase, stabilizes through mitochondrial membrane depolarization and activates Parkin (or Park2) to trigger mitophagy [75]. Defective of Parkin/Pink1signaling affects the clearance of damaged mitochondria in HSCs which ultimately impairs the self-renewal of highly potent HSCs. In relation, deletion of AAA+-ATPase (*Atad3a*), a regulator of Pink1-dependent mitophagy, hyperactivated mitophagy and expanded the number of HSCs.

The activation of Pink/Parkin1 and accumulation of mitophagy receptor proteins distinguish damaged mitochondria from undamaged ones. While drugs causing depolarization of the mitochondrial outer membrane or mtDNA mutations cause mitophagy, factors which stimulate mitophagy or autophagy under physiological conditions are unclear. Mediators of HSC maintenance such as Foxo3a have been reported to regulate mitochondria functions along with the expression of stress-induced autophagy genes, autophagy related 4b (*Atg4b*), microtubule associated protein 1 light chain 3β (*Map1lc3b*), and BCL2 interacting protein 3 (*Bnip3*) [76]. Likewise, mitochondria volume and autophagy alteration in dysfunctional HSCs is noted in a variety of studies focusing on metabolic regulators. Other than the genetic modulation of metabolic programs, mitochondria damage precipitates through the changes in mitochondria metabolites such as ROS or calcium [77,78]. Nonetheless, the full characterization of damaged mitochondria in stressed and physiological HSCs as well as in depth understanding of the mechanism of damage is lacking.

## 7. The Role of Lysosome and Mitochondria Network in HSCs

Mitophagy requires the imminent fusion of autophagosomes with lysosomes for the enzymatic degradation of damaged mitochondria. The regulation of lysosomal volume and activation have recently been highlighted in HSC regulation. ΔΨmt low HSCs which have high long-term reconstitution capacity were enriched in lysosomal- and proteasomal-mediated pathway gene expression [79]. Interestingly, highly potent, quiescent HSCs presented with higher volumes of large inactive lysosomes compared to cycling HSCs. The repression of lysosome function with concanamycin A (ConA), lead to a further increase in lysosomal volume, enhanced mitochondria sequestration in lysosomes as well as decreased glycolysis activity. In a single-cell based observation of HSC divisions, lysosomes as well as autophagosomes and mitophagosomes were asymmetrically co-inherited to daughter cells [80]. The disproportional inheritance of organelles affected the metabolic state (ΔΨmt and ROS) and dictated the differentiation fate of daughter cells.

While the inheritance and presence of lysosomes for mitochondria clearance inarguably associates with HSC fate, less is known about whether and how mitochondria and lysosomes interact and form an inter-organelle network for HSC regulation (Figure 2). It has been proposed that organelles within the cytoplasm coordinate a network and can relay retrograde signals to the nucleus to control cellular homeostasis [81]. Rather than being degraded, mitochondria may interact with lysosomes for cross-organelle signaling. The GTPase, Ras related in brain 7 (Rab7), promotes tethering of mitochondria to lysosomes leaving contacts mark sites which facilitate mitochondrial fission in cultured cell lines [82]. Therefore, lysosomes may mediate mitochondrial fission independent of autophagy. In addition, mitochondria and lysosome contacts facilitate the inter-organelle flux of metabolites such as lipids, calcium ions and iron ions [83]. In particular, both mitochondria and lysosomes are major storage sites of calcium and iron ions which play vital roles in cellular homeostasis. Retrograde signaling through organelles involve a complex network between mitochondria and lysosomes along with other organelles such as the endoplasmic reticulum. Whether and to what degree physiological HSCs rely on inter-organelle crosstalk are yet to be confirmed.

## 8. Regulation of Cellular Metabolism through Folliculin Signal

Mitochondria and lysosomes share critical molecular pathways for their biogenesis and activation [81]. Transcriptional activation of cyclic adenosine monophosphate (cAMP) response element binding protein (CREB), FoxO, E2F1, TFEB and TFE3 have been implicated in the activation of both mitochondrial and lysosomal biogenesis as well as regulation of autophagy. Notably, the tumor suppressor Flcn contributes to maintain the homeostasis of lysosomes, mitochondria and autophagic processes. Flcn is a multifunctional molecule affecting diverse signaling pathways, including mechanistic target of rapamycin complex 1 (mTORC1)/AMP-activated protein kinase (AMPK) pathway, TFE3/TFEB and peroxisome proliferator-activated receptor gamma coactivator 1α (PGC-1α)/transcription factor A, mitochondrial (TFAM) [84]. Germ line mutations in *FLCN* are responsible for the autosomal dominant inherited disorder Birt–Hogg–Dubé (BHD) syndrome which presents lung and kidney cysts and kidney cancer [85]. Overexpression of *Tfe3* in mice led to the formation of kidney cysts and cancer while the genetic depletion of *Tfeb* rescued the phenotype of mice with kidney-specific knockout of *Flcn*. Therefore, it is likely that the activation of *TFEB* is the main driver of the kidney pathology in BHD syndrome [86].

The multi-functional nature of FLCN insinuates it as a key regulator in lysosomal and mitochondrial dynamics. FLCN and its binding proteins folliculin interaction protein 1 (FNIP1) and folliulin interaction protein 2 (FNIP2) form a stable complex (lysosomal folliculin complex (LFC)) with Rag GTPases and Ragulator on the lysosomal membrane during starvation. GAP (GTPase activating protein) activity of FLCN, which switches inactive RagC/D (RagC/D^GTP^) to its active form (RagC/D^GDP^) through nucleotide hydrolysis, is restrained upon complex formation. Stabilization of LFC promotes mTORC1 release to the cytosol, maximizes nuclear localization of the TFE3/TFEB and promotes autophagy. Conversely, in high nutrient conditions, LFC disassembles and allows FLCN to convert RagC/D^GTP^ to RagC/D^GDP^, which results in the recruitment of mTORC1 to the lysosomal membrane and phosphorylation of TFE3/TFEB [87] (Figure 3A). Nuclear translocation of TFE3/TFEB is thus prevented by Flcn-mediated activation of RagC/D which stations TFE3/TFEB on the lysosome surface and activated mTORC1 [88]. *Flcn*-deficiency or mutation causes aberrant TFE3/TFEB transcription and mTORC1 activity. Loss of FLCN function hyperactivates TFE3/TFEB transcription of lysosome-associated genes, including RagC/D. Enhanced transcription of RagC/D leads to an increase of its inactive form (RagC/D^GTP^) which cannot tether TFE3/TFEB to the lysosomes yet activates mTORC1 and its downstream signal eukaryotic translation initiation factor 4E binding protein (4EBP) and S6 Kinase (S6K) (Figure 3B). mTORC1 is thus consistently activated although RagC/D GTPases are inactive in the absence of FLCN. As a net result, cells with Flcn-deficiency inadequately sense and respond to nutrient availability [86]. In relation, the function of mTORC1 on lysosomal surfaces as a critical metabolic regulator to govern amino acid efflux and content within lysosomes was reported [89]. Inhibition of mTORC1, and not the vacuolar H^+^-adenosine triphosphatase (V-ATPase), reduced the lysosomal efflux of essential amino acid, converting the lysosome into a cellular depot of amino acids.

FLCN is essential for the maintenance of hematopoiesis in the BM. Hematopoietic cell-specific deletion of *Flcn* (*Flcnfl/fl;MxCre1+* mice) hyper-activates phagocytic macrophages through alterations in metabolism. Flcn-deficiency stimulates the nuclear localization and transcriptional activity of TFE3 and lysosomal biogenesis which leads to aberrant expansion phagocytic macrophages in bone marrow [90]. Increase in the transcriptional activity of TFE3 accelerated gamma aminobutyric acid A receptor associated protein (GABARAP: an Atg/8LC3 family member)-mediated formation of autophagosomes and increased the expression of lysosomal genes [91]. *Flcn*-deletion in murine hematopoietic cells drives BM HSCs into proliferative exhaustion which results in acute bone marrow failure. HSCs from *Flcn*-deficient mice lose quiescence and exhibit abnormal cell cycle progression leading to apoptosis [92]. While mTORC1 regulates HSC quiescence and proliferation [93,94], HSCs are unimpaired upon RagA GTPase inhibition and upregulate mTORC1 in a RagA- and nutrient-independent manner [95]. It is not clear how Flcn-deficiency affects HSC metabolism and whether alterations in the Flcn pathway affect mTORC1 function and nutrient sensing in HSCs.

FLCN not only regulates autophagy and lysosome biogenesis but also mitochondria biogenesis [84]. Flcn-deficiency in myocytes upregulated mitochondrial biogenesis and OXPHOS through the activation of the transcription factor PGC-1α [96]. Furthermore, the degradation of FNIP2 in myoblasts was stimulated through reductive stress and resulted in the modulation of mitochondria OXHPOS activity and ROS production [97]. Collectively, Flcn significantly affects mitochondria, lysosomes and oxidative stress and its precise role in cell organelle collaboration during lifelong stem cell regulation may uncover a new field in stem cell biology.

## 9. Conclusions

The determination of HSC cell fate is the summation of cell intrinsic and extrinsic regulators which alter genetic, epigenetic and metabolic states. While the hypoxia and oxidative stress settle as the main regulatory elements of HSC metabolism, recent research has consolidated the involvement of organelles for the determination of HSC cell fate. However, current studies still lack in precision analysis and insight when it comes to how these organelles regulate HSCs. Which organelles and how does their non-bioenergetic function associate with HSC regulation? Does hypoxia, redox state or nutrient sensing of HSCs affect the morphology and dynamics of organelles? Which molecular pathway controls inter-organelle networks for the homeostasis of HSCs? Metabolic, genomic and epigenomic analysis of Flcn-deficient HSCs at a single cell level may identify pathways which serve to maintain organelle homeostasis in HSCs. Furthermore, detailed 3-dimenstional ultrastructural imaging of organelles within HSCs along with time-lapse imaging of organelles in cultured HSCs may uncover novel molecules which regulate the dynamics of organelle networks in HSCs. Further analysis on HSCs, especially at a single cell level, should answer these questions and pave the way to maintain healthy lifelong hematopoiesis.

## Figures and Tables

**Figure 1 ijms-22-04627-f001:**
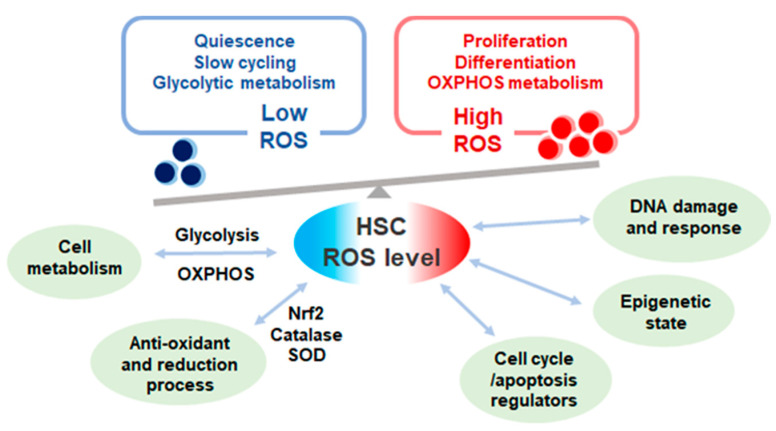
The levels and effect of reactive oxygen species (ROS) in quiescent and proliferative hematopoietic stem cells (HSCs). ROS level in HSCs varies according to cell cycle state. Quiescent HSCs depend on glycolytic metabolism exhibit low ROS level while proliferative HSCs depend on oxidative phosphorylation (OXPHOS) metabolism with high ROS production. ROS levels affect multiple cell functions in HSCs and heavily influences HSC fate.

**Figure 2 ijms-22-04627-f002:**
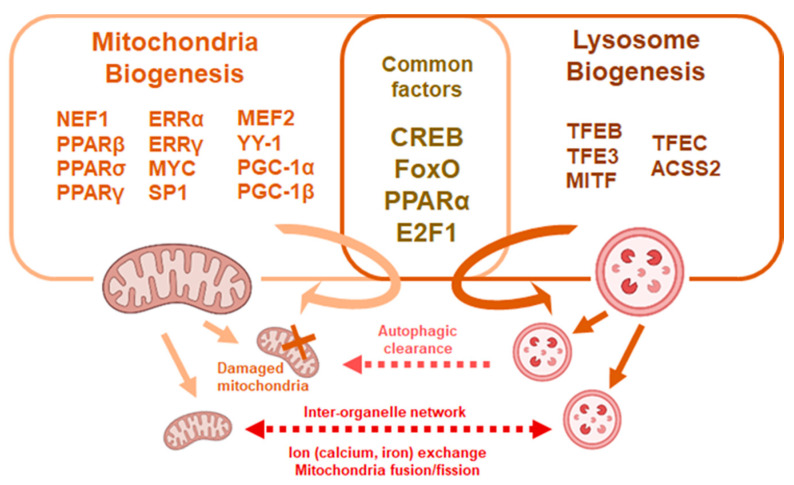
Common factors affect mitochondria and lysosome biogenesis. Mitochondria and lysosomal biogenesis are regulated through multiple factors, some of which are common to both processes. Mitochondria and lysosomes crosstalk though fusion and exchange of metabolites or through autophagic processes.

**Figure 3 ijms-22-04627-f003:**
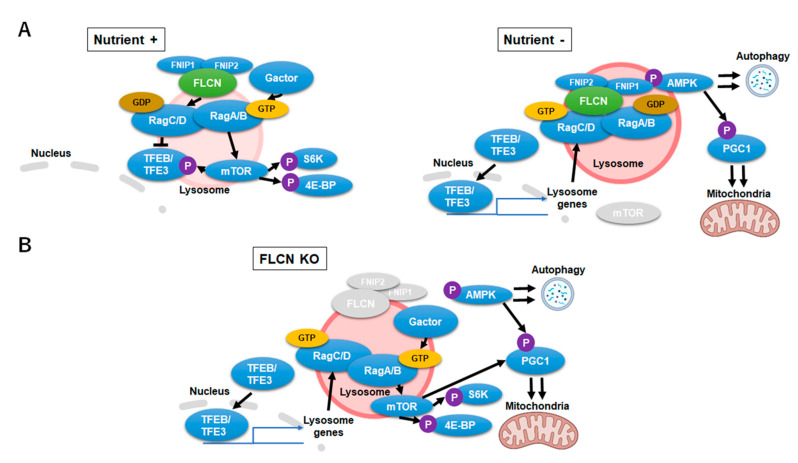
Folliculin-Transcription Factor EB/Transcription Factor Binding To IGHM Enhancer 3 (FLCN-TFEB/TFE3) signaling during nutrient-rich, starvation and FLCN-deficient conditions. (**A**) FLCN-TFEB/TFE3 signaling during nutrient-rich (left) and starvation (right) conditions. Upon starvation, FLCN (LFC) stabilizes on the lysosomal surface which promotes TFEB/TFE3 transcription. (**B**) Signaling alternations during FLCN-deficiency. FLCN-deficiency causes upregulation of the inactive RagC/D^GTP^ which leads to an aberrant activation of TFE3/TFEB and mechanistic target of rapamycin complex 1 (mTORC1) signaling.

## Data Availability

Not applicable.

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
