# Peer review of "Mitochondria Turnover and Lysosomal Function in Hematopoietic Stem Cell Metabolism"

_ijms, 2021, doi:10.3390/ijms22094627_

Round 1

Reviewer 1 Report

This review article by Mochizuki-Kashio and colleagues focuses on mitochondrial turnover and lysomal function in hematopoietic stem cell (HSC) metabolism. They are leading experts in the field, and correspondingly the review is well-written, up-to-date, and fairly comprehensive. This reviewer just has a few suggestions that might improve it prior to publication:

  1. It would be most useful to readers if the official gene names are included throughout (e.g. Risp and others).
  2. A statement within the introduction about the major focus of this review on mouse HSCs would be useful to avoid reader confusion.
  3. This is certainly not the first review to take on HSC metabolism and several of the authors own reviews are cited. Given the focus of this review on mitochondrial metabolism, it would be useful to direct readers to other more comprehensive reviews by themselves as well as others (e.g. Ito and Ito, Experimental Hematology 2018, Filippi, 2021, and others).
  4. The authors imply that the major target of the TF Mesi1 is Hif1a on line 46. Is this correct? Perhaps the authors can better qualify this statement.
  5. The authors discuss maintenance of HSCs in ex vivo hypoxic conditions in line 50. However, recent long-term HSC expansion protocols (e.g. Wilkinson et al., Nature 2019) were performed at 20% O2. Can the authors comment on this potential discrepancy?
  6. Given the focus on mitochondria, this review could be improved with more discussion of mitochondrial DNA and its potential role(s) in HSC function.
  7. Mitochondria are also a major hub for amino acid metabolism but this is not discussed besides glutaminolysis. This review would benefit from some more discussion in this area.
  8. The authors suggest potential questions for the field to focus on in the Conclusion but don’t give many ideas on how this might be performed. Could the authors elaborate further? Are there technical advances that would help the field move forwards?

Reviewer 2 Report

In the manuscript titled “Mitochondria turnover and lysosomal function in hematopoietic stem cell metabolism”, Mochizuki-Kashio et al reviewed literatures that reveal critical roles of mitochondria and lysosomes in hematopoietic stem cell (HSC) maintenance. This is an emerging field with a growing body of recent findings to support the authors’ central thesis, that these two organelles regulate multiple metabolic and signaling pathways to determine and alter HSC fate. Overall, the manuscript is informative and compelling. I only have the following comments/suggestions for the authors to address:

  1. Line 20-21: HSC is a single cell type but not a single cell.
  2. Line 36: The word “overlooking” is incorrect. It means neglecting.
  3. Line 49-52: What is the literature reference for these culture conditions?
  4. Although it is stated that “Stem cells rely on glycolysis for energy production…”, HSCs seems to rely on OXPHOS to produce ATP [Line 69-71], while consuming low amount of glucose [Line 90]. This sets up a conundrum as “HSCs… utilize anerobic glycolysis as their main energy source” [Line 53-54]. Can the authors discuss and contrast these statements? Has any study reported which pathway the HSCs are more dependent on for ATP production?
  5. Line 109: The word “acquiesce” personifies HSCs and should be replaced.
  6. Line 194-195: What is the literature reference that showed HSC mitochondria are inactive in producing energy?
  7. Line 227: Ref 71 is not relevant to Atad3a and mitophagy in HSCs.
  8. Line 243: Lysosomes use acid hydrolases to degrade macromolecules. “Enzymatic degradation” should be used.
  9. Line 245-246: Either “enriched in … gene expression” or “express high levels of … genes”.
  10. Line 260: There is no evidence of mitochondria “fusing” with lysosomes. “Interact” or “form membrane contacts” is more appropriate.
  11. Line 278-280: Please provide references related to these TFs.
  12. Line 290-339: It is unclear to me why FLCN was emphasized in the manuscript. After all, only one reference (Ref 87) was related to HSCs and mentioned in the paragraphs. The other pivotal lysosomal signaling pathway, mTORC1, has been shown by multiple groups to be critical to HSC quiescence [Kalaitzidis et al JCI 2017; Gan et al Cell Cycle 2009; Fan et al PNAS 2021; etc]. These studies should be mentioned and discussed in the lysosome section of the manuscript.
